# The Impact of Vegan Diet in the Prevention and Treatment of Type 2 Diabetes: A Systematic Review

**DOI:** 10.3390/nu13062123

**Published:** 2021-06-21

**Authors:** Daniela Pollakova, Aikaterini Andreadi, Francesca Pacifici, David Della-Morte, Davide Lauro, Claudio Tubili

**Affiliations:** 1Diabetes Unit, San Camillo Forlanini Hospital, Circonvallazione Gianicolense 87, 00152 Rome, Italy; ctubili@scamilloforlanini.rm.it; 2Department of Systems Medicine, School of Medicine and Surgery, University of Rome Tor Vergata, Via Montpellier 1, 00133 Rome, Italy; andreadi@med.uniroma2.it (A.A.); pacifici.francesca@gmail.com (F.P.); david.dellamorte@uniroma2.it (D.D.-M.); d.lauro@med.uniroma2.it (D.L.); 3Unit of Endocrinology and Diabetes, University Hospital, Fondazione Policlinico Tor Vergata, Viale Oxford 81, 00133 Rome, Italy

**Keywords:** vegan diet, plant-based diet, type 2 diabetes, insulin resistance

## Abstract

A protective effect of vegan diets on health outcomes has been observed in previous studies, but its impact on diabetes is still debated. The aim of this review is to assess the relationship between vegan diets and the risk for type 2 diabetes (T2D) along with its effect on glycemic control and diabetes-related complications. In accordance with PRISMA (Preferred Reporting Items for Systematic Reviews and Meta–Analyses) guidelines, Pubmed and Cochrane library databases were systematically searched for all relevant studies. Seven observational and eight randomized controlled (RCTs) studies were included. The methodological quality of studies was assessed using the National Institutes of Health quality assessment tool for observational cohort and cross-sectional studies and the Cochrane Risk of Bias Tool for RCTs. We found that a vegan diet is associated with lower T2D prevalence or incidence and in T2D patients decreases high glucose values and improves glucose homeostasis, as reported from the majority of included studies. This approach seems to be comparable to other recommended healthful eating models, but as it may have potential adverse effects associated with the long-term exclusion of some nutrients, appropriate nutritional planning and surveillance are recommended, particularly in specific groups of diabetic patients such as frail elderly, adolescents, and pregnant or breastfeeding women.

## 1. Introduction

Interest in vegan diets is increasing around the world: it is estimated that the number of vegans in the US grew by 600% from nearly 4 million in 2014 to 19.6 million in 2017 [1]. Ideological, environmental, ethical, political or religious arguments are the most important reasons for this rapid growth.

An association between a vegan diet and a reduced risk of total cancer incidence has been demonstrated in large prospective cohort studies [2], but its impact in diabetes mellitus (DM) is still under debate. As diet and lifestyle are the fundamental pillars of DM prevention and therapy and plant-based diets (PBD) are considered an example of healthful eating patterns and are recommended for individuals with DM [3], we may also expect some beneficial effects in the case of the vegan diet.

The aim of this paper is to conduct a systematic review of all observational and intervention studies that describe the relationship between vegan diets and the risk for type 2 diabetes (T2D), along with its effect on glycemic control and T2D-related complications.

## 2. Materials and Methods 

Following the PRISMA (Preferred Reporting Items for Systematic Reviews and Meta–Analyses) statement [4] we systematically conducted electronic searches in Pubmed and Cochrane library databases including all studies published in English language until 23 May 2020. Additional manual searches using the reference lists of identified articles, reviews and meta–analyses were also performed. The search strategy included the following keywords used in all fields or in combination as MeSH (Medical Subject Headings) terms: “veganism”/”vegan diet”/”vegans”/”vegan dietary pattern”/”plant–based diet”/”plant–based dietary pattern” and “diabetes”/”diabetes mellitus”/”type 2 diabetes”/”type 2 diabetes mellitus”.

We considered all observational (i.e., cross–sectional or prospective cohort studies) and randomized controlled studies based on the PICO framework (P—population: presumably healthy adults in case of observational studies or individuals with T2D ≥ 18 years in case of randomized controlled studies; I—intervention: vegan or plant–based diet, defined as a dietary pattern that omit all the animal–derived products; C—comparisons: omnivorous diet, defined as a consumption of all types of foods, including meat, meat products and poultry; O—outcomes: prevalence or incidence of T2D in case of observational studies, glycemic control—fasting glucose and Hemoglobin A_1c_ (HbA_1c_) levels and risk factors for diabetes related complications—Body Mass Index (BMI), waist circumference, total cholesterol, low-density lipoprotein (LDL) cholesterol, high-density lipoprotein (HDL) cholesterol, triglycerides, systolic and diastolic blood pressure, microalbuminuria or other parameters of renal function, electrochemical skin conductance or other parameters of nerve function in case of randomized controlled studies. Studies were excluded if there was no vegan intervention (e.g., flexitarian, pesco-vegetarian, lacto-ovo-vegetarian); intervention < 4 week; no omnivorous control; were not conducted in humans or in individuals with T2D (e.g., prediabetes or type 1 diabetes); were conducted in children, adolescents, pregnant or breastfeeding women. When multiple articles for a single study were present, we considered the most recent publication, eventually completed with data from the previous reports. Three investigators (Daniela Pollakova, Davide Lauro and Claudio Tubili) independently reviewed and extracted relevant data. All discrepancies were resolved by consensus.

### Methodological Quality Assessment of Studies

The included studies were independently assessed by three reviewers (Daniela Pollakova, Davide Lauro and Claudio Tubili) for methodological quality using elements of the National Institutes of Health (NIH) quality assessment tool for observational cohort and cross-sectional studies [5] and the Cochrane Risk of Bias Tool [6] in the case of randomized controlled studies. In the NIH, possible sources of bias (e.g., selection, information, performance, attrition, measurement/detection), confounding, reporting and other factors were assessed through 14 items. Studies were rated as good (low risk of bias), fair (study susceptible to some bias) and poor (significant risk of bias) quality. In randomized controlled studies, an assessment of the risk of bias was carried out across 5 domains (sequence generation, allocation concealment, blinding, incomplete outcome data and selective outcome reporting). Studies were identified as being of low risk if proper methods were taken to reduce bias (1 point), of high risk if improper methods creating bias were taken (0 point) or of unclear risk if insufficient information was provided to determine the bias level (0.5 point). A total possible score of 5 points could be obtained; studies scoring 5.0–4.5 were considered as being of low risk, 4.0–3.5 points as of moderate risk and <3.5 points as of high risk of bias. All disagreements were discussed and resolved.

## 3. Results

The initial search identified 898 results (Pubmed = 824, Cochrane Library = 66, other sources = 8) and after the application of inclusion/exclusion criteria, a final sample of 15 articles (7 observational and 8 randomized controlled studies) were included for this review (Figure 1).

### 3.1. Observational Studies

The main characteristics of the included observational studies reporting the effects of vegan or plant-based dietary patterns on risk for T2D (prevalence or incidence) are summarized in Table 1. Of the seven studies, five were classified as prospective cohort studies [7,8,9,10,11] and two as cross-sectional studies [12,13]. The study quality assessment for each observational study reported a good quality in four studies [9,10,11,13] and a fair quality in three studies [7,8,12]. No study was rated of poor quality.

### 3.2. Studies Evaluating Prevalence of Type 2 Diabetes

Out of the seven observational studies, two cross-sectional studies reported the prevalence of T2D among vegans [12,13]. In the Adventist Health study-2 [12] that included more than 60,000 subjects (22,434 men and 38,469 women) from US and Canada and aged 30 years or older, the prevalence of T2D was higher among nonvegetarians (7,6%) than in the various types of vegetarianism (semi-vegetarians 6,1%; pesco-vegetarians 4,8%; lacto-ovo-vegetarians 3,2%; vegans 2,9%) and among these, the lowest prevalence of T2D was seen in vegans (2,6x lower than in omnivores). It is worth noting that the difference in the mean BMI between vegans and omnivores was more than 5 units (23.6 kg/m^2^ vs. 28.8 kg/m2, respectively), but also after adjustment for BMI and others variables (age, sex, ethnicity, education, income, physical activity, television watching, sleep habits, alcohol use), vegans had a lower risk of T2D than nonvegetarians (Odds Ratio (OR) 0.51, 95% Confidence Interval (CI) 0.40–0.66). Different results were observed in India’s third National Family Health Survey [13]. In this cross-sectional study of 156,317 participants (56,742 men and 99,574 women) aged 20–49 years, no significantly different prevalence of T2D between vegans (*n* = 2560) and omnivores (*n* = 99,372) was observed (1.0% vs. 1.2%, respectively) and the difference between the mean BMI was also low (20.5 kg/m^2^ in vegans vs. 20.7 kg/m^2^ in omnivores).

### 3.3. Studies Evaluating Incidence of Type 2 Diabetes

From the seven observational studies, five prospective cohort studies evaluated the incidence rate of T2D among vegans [7,11] or people that consumed a PBD [8,9,10].

In the population of the Adventist Health Study-2 [7] that included more than 40,000 participants (15,200 men and 26,187 women) free of DM at baseline, cases of T2D developed in 0.54% of vegans (*n* = 3545), respective to 2.12% of omnivores (*n* = 17,695) after 2 years. The vegan diet was resulted as protective for the development of T2D, respective to non-vegetarians in both non-Black (White non-Hispanic, Hispanic, Middle Eastern, Asian, Native Hawaiian/other Pacific Islander or American Indian) and Black (African American, West Indian/Caribbean, African or other Black) participants (OR 0.429, 95% CI 0.249–0.740 and OR 0.381, 95% CI 0.236–0.617, respectively) after adjustment for age, sex, education, income, television watching, physical activity, sleep, alcohol use, smoking and BMI.

Slightly different results were observed in the EPIC–Oxford cohort study of 45,314 UK adults (10,737 men and 34,577 women) aged 20–90 years. Over a mean follow-up of 17.6 years, the percentage of incident T2D cases among 1781 vegans was lower than in 22,796 omnivores (1.45% vs. 3.83%, respectively), and after adjustment for age, education, socio-economic status, physical activity, ethnicity, smoke and alcohol intake, the hazard ratio for developing T2D in vegans was significantly lower than in regular meat eaters (HR 0.53, 95% CI 0.36–0.79). However, when the analysis was further adjusted for BMI, the significance was lost (Hazard Ratio (HR) 0.99, 95% CI 0.66–1.48). It is important to notice that authors have evaluated regular meat eaters (defined as subjects who consumed ≥ 50 g of meat per day) and low meat eaters (consumption of <50 g of meat per day) separately, using regular meat eaters as a reference group but not a whole sample of omnivores.

In the ATTICA Cohort Study [9], the dietary habits of 1485 Greek adults (726 men and 759 women) aged 18–89 years and free of DM and any cardiovascular disease at baseline were evaluated. After 10 years of follow up, a crude T2D incidence rate of 12.9% was observed. Using the a posteriori method for assessing the most healthful dietary pattern and after stratification by age-group, as well as after adjustment for sex, family history of DM, waist circumference and smoking status, it was observed that the consumption of fruits, vegetables, legumes, bread, rusk and pasta (all food groups “allowed” to be consumed by vegans) reduced the 10-year T2D risk by 40%, but only in the 45–55 years age group at baseline with marginal statistical significance (OR 0.60, 95% CI 0.34–1.07). When the analysis was additionally adjusted for the percentage of calories from carbohydrates and for total energy intake, statistical significance was lost completely (OR 0.62, 95% CI 0.34–1.13 and OR 0.72, 95% CI 0.35–1.49, respectively), indicating a partially mediating effect of carbohydrates and the importance of the total amount of assumed calories. Not all studies dichotomously classified dietary patterns as vegan or plant-based and omnivorous. Several studies investigated the variation in the degree of having a plant-based versus animal-based diet.

The impact of PBD, and its healthy and unhealthy version, on T2D incidence was studied on the sample of more than 200,000 participants (40,539 men and 160,188 women) aged 25–75 years, recruited in three US cohort studies: Nurses’ Health Study, Nurses’ Health Study 2 and Health Professionals Follow-up Study [8]. All participants were free of any chronic disease at baseline and were followed for more than 20 years. In these studies, there was 16,162 incident cases of T2D in 4,102,369 person–years of follow-up. Authors created an overall “Plant-based Diet Index” (PDI)—the more plant foods were consumed, a higher score was obtained, while the consumption of animal foods lowered the score. The PDI was then divided in deciles. After adjustment for BMI, age, smoking status, physical activity, alcohol intake, multivitamin use, family history of DM, margarine and energy intake, baseline hypertension and baseline hypercholesterolemia, it was observed that subjects with the highest decile of PDI compared to those in the lowest one had a reduction of about 20% in the risk of T2D (HR for extreme deciles 0.80, 95% CI 0.74–0.87). Then, healthful and unhealthful versions of PDI were created (hPDI and uPDI, respectively) with the same modalities; for instance, vegetables, fruits, legumes, whole grains or vegetable oils were considered healthy PDI; on the other hand, refined grains, fruit juices, sweetened beverages, potatoes, desserts, etc. were calculated as unhealthy PDI. Evaluating the difference between the highest and the lowest decile of hPDI, there was a further 34% reduction in T2D disease (HR 0.66, 95% CI 0.61–0.72). On the contrary, the difference in uPDI resulted in the increased risk of about 16% (HR 1.16, 95% CI 1.08–1.25).

Additionally, in the Rotterdam Study (RS) [10], a prospective population-based cohort study, the continuous PDI was constructed (range 0–92) in a population of 6798 participants (2808 men and 3990 women), with an age of 62.7 ± 7.8 years. Food-frequency questionnaires at baseline of three RS groups (RS-I-1: 1989–1993, RS-II-1: 2000–2001, RSIII-1: 2006–2008) were collected. After a median follow-up of 7.3 years and withdrawal of 28 subjects, during 54,024 person–years of follow-up, amongst 6670 subjects, 642 incident cases of T2D were documented. After adjustment for energy intake, sex, age, RS sub-cohort, smoking status, educational level, physical activity, dietary supplement use and family history of DM, a higher score on the PDI was associated with a lower diabetes incidence (per 10 units higher score on index HR 0.82, 95% CI 0.73–0.92) and also after adjustment for BMI, the significance was maintained (HR 0.87, 95% CI 0.79–0.99).

### 3.4. Randomized Controlled Studies

Eight randomized controlled studies evaluating the impact of vegan or PBD on glycemic control and diabetes-related complications in T2D patients met all inclusion criteria. The main characteristics of the included studies are summarized in Table 2. The risk of bias assessment for each study reported a high risk of bias in three studies [14,15,16] and a moderate risk in five [17,18,19,20,21] studies. No study was rated to have a low risk of bias, mainly due to impossible double-blinding in dietary intervention trials.

### 3.5. Studies Evaluating Glycemic Control

All included studies reported a reduction between −0.3% [17] to −1.4 % [14] of HbA_1c_ levels after vegan diet, and in five studies, this reduction reached statistical significance [15,18,19,20,21]. On the other hand, in control groups, six studies reported a reduction in HbA_1c_ levels [14,15,16,17,20,21] (three reached the statistical significance [15,20,21]) that varied from −0.14% [16] to −1.0% [14]. Bunner et al. [19] reported no change from baseline and Mishra et al. [18] described the increase by 0.1% and only in two studies [18,20], the mean between-group difference of changes from baseline to final values was significant (Table 3).

Not all studies reported the effect of the vegan diet on fasting plasma glucose (FPG). Out of six studies evaluating this variable [14,15,16,19,20,21], only Nicholson et al. [14] reported a significant difference between groups assessing the intervention mean created by averaging the six biweekly measures with a reduction of 28% from baseline to the intervention mean in the vegan group, compared to a 12% decrease in the control group. Barnard et al. [16] reported a significant decline only in the intervention group. In other studies, the reduction, although not significant, was observed in both the intervention and the control groups, except for the study of Lee et al. [20] where the values of fasting plasma glucose remained unchanged in the control group (Appendix A).

### 3.6. Studies Evaluating Risk Factors for Diabetes Related Complications

Five studies reported the BMI measurement as an outcome [16,18,19,20,21] and all of them noticed its significant reduction after the vegan diet that varied from −0.4 kg/m2 [20] to −2.4 kg/m^2^ [19]. On the other hand, in control groups, four studies reported a reduction in BMI [16,19,20,21] (two of them [16,21] reached significance) that varied from −0.1 kg/m^2^ [20] to −1.5 kg/m^2^ [21] and Mishra et al. [18] reported no change from baseline. Only in two studies [18,19] was the mean between-group difference of changes from baseline to final values significant.

As regards the waist circumference, three studies evaluated this outcome [16,17,20], and all of them reported its significant reduction in the intervention group that varied from −3.1 cm [20] to −4.7 cm [17]. In the control groups, Barnard et al. [16] reported a significant reduction by −1.8 cm, Lee et al. [20] a not significant reduction by −0.8 cm and Ferdowsian et al. [17] described a not significant increase by 0.8 cm. Two studies [17,20] reported a significant mean between-group difference.

All studies reported the evaluation of systolic (SBP) and diastolic (DBP) blood pressure. In the intervention groups, the changes in SBP varied from −11.5 mmHg [19] to +1.0 mmHg [20], but only in the study of Mishra et al. [18] did this change reach significance. Generally, five studies reported the reduction [14,15,18,19,21], two studies reported no variation [16,17] and only Lee et al. [20] reported the increase in SBP. In the control groups, six studies [14,15,18,19,20,21] reported the reduction and two studies [16,17] the increase in SBP from baseline to final analysis that varied from −18.9 mmHg [14] to +5.7 mmHg [17], but only in two studies [17,18] did this difference reach statistical significance and Ferdowsian et al. [17] alone reported a significant mean between-group difference. As for the DBP, in the intervention groups, only Lee et al. [20] reported a not significant increase in DBP by +1.1 mmHg. In the other studies, its reduction was described, that varied from −5.8 mmHg [14] to −0.4 mmHg [17], and in four studies [15,16,18,19] these changes were significant. In the control groups, only Ferdowsian et al. [17] reported a significant increase in DBP by +5.1 mmHg; in the other studies, the described DBP variation was between −10.6 mmHg [14] to −1.4 mmHg [20] and three of them were significant [15,16,18]. Again, only Ferdowsian et al. [17] reported a significant mean between-group difference in DBP values.

All studies except Lee et al. [20] reported changes in total cholesterol, and all of them noticed its reduction after both the intervention and the control diet, omitting the control group in the study of Bunner et al. [19] (increase by 2.2 mg/dL), and this reduction varied from −24.32 mg/dL [14] to −9.8 mg/dL [17] after the vegan diet (four reached statistical significance [15,16,17,18]) and from −24.32 mg/dL [14] to −1.3 mg/dL [18] in the control groups (three were significant [15,16,21]). Only Mishra et al. [18] noticed a significant in-between group difference of changes from baseline to final values.

All studies reported the evaluation of HDL cholesterol. In the intervention groups, the changes from baseline to final values varied from −7.72 mg/dL [14] to + 2.2 mg/dL [20] and in five studies, these changes reached the significance [14,17,18,19,21]. Generally, six studies reported the reduction [14,16,17,18,19,21] and two studies [15,20] reported the increase in HDL cholesterol. Additionally, in the control groups, six studies [14,15,16,17,19,21] described its reduction (from −2.4 mg/dL [19] to −0.4 mg/dL [17]), in two studies [18,20] HDL cholesterol increased (by + 0.5 mg/dL [20] and + 0.7 mg/dL [18]) and none of these changes were statistically significant. Regarding the mean in-between group difference, only in the studies of Ferdowsian et al. [17] and Mishra et al. [18] did HDL cholesterol values reach statistical significance.

On the other hand, not all studies evaluated changes in LDL cholesterol [16,17,18,19,20,21]. Only Bunner et al. [19] reported its slight increase in the control group by 0.4 mg/dL; in other studies, a reduction was observed in both intervention (varying from −13.5 mg/dL [16] to −2.8 mg/dL [20]) and control groups (from −12.7 mg/dL [21] to −1.0 mg/dL [20]). In intervention groups, the statistical significance was reached in three studies [16,18,21] and in control groups in two studies [16,21], with the mean in-between group difference significant in the study of Mishra et al. [18].

Relatively uneven results were observed assessing triglyceride levels; all studies reported changes from baseline to final values, expressed in mg/dL. Only Wheeler et al. [15] reported the change as the Area Under Curve (AUC). In the intervention groups, four studies noticed a reduction [14,15,16,17] (varying from −33.9 mg/dL [16] to −4.4 mg/dL [17]) and four an increase [18,19,20,21] (from + 4.7 mg/dL [19] to + 20.8 mg/dL [21]) in triglyceride levels, and only in two studies were these changes significant [16,18]. In the control groups, six studies reported a decrease [14,15,16,18,20,21] (from −38.93 mg/dL [14] to −2.9 mg/dL [18]) and two studies an increase (by + 3.5 mg/dL [17] and + 21.9 mg/dL [19]) in this variable, and significance was only observed in the study of Barnard et al. [21]. Only in the study of Mishra et al. [18] was the mean in-between group statistical significance reached, although Lee et al. [20] observed a marginal significance.

It is worth saying that in the study of Ferdowsian et al. [17] and Mishra et al. [18], not all participants, in whom the cardiometabolic risk factors mentioned above were measured, had DM.

Evaluating renal function, four studies reported diverse outcomes: Nicholson et al. [14] and Barnard et al. [16] evaluated the 24 h microalbuminuria. A decrease was observed in the intervention groups from 434.3 mg/24 h to 155.2 mg/24 h and from 33 mg/24 h to 20.2 mg/24 h in the study of Nicholson et al. [14] and in the study of Barnard et al. [16], respectively. Conversely, an increase (from 82.9 mg/24 h to 169.2 mg/24 h in the study of Nicholson et al. [14] and from 55 mg/24 h to 69.5 mg/24 h in the study of Barnard et al. [16]) was noticed in the control groups; however, none of these changes reached statistical significance (for a wide range of baseline values in the study of Nicholson et al. [14]), but the mean in-between group difference was resulted as marginally significant in the study of Barnard et al. [16]. Additionally, in the study of Barnard et al. [21], the 24 h albuminuria was assessed, but the authors expressed this outcome as median mg/dL. Differently from previous studies, a significant increase in albuminuria was observed in the intervention group, while in the control group values, dropped insignificantly, with a significant mean in-between group difference. Wheeler et al. [15] measured other parameters: Albumin Excretion Rate (AER), Glomerular Filtration Rate (GFR) and Renal Plasma Flow (RPF). However, there were not any significant differences in the change from baseline to 6 weeks or between the intervention and the control groups.

Changes in clinical outcomes from baseline to final values by group assignment are summarized in Appendix A.

Finally, Bunner et al. [19] evaluated the impact of the vegan diet on pain in chronic diabetic neuropathy. Different questionnaires or scales assessing the painful and sensory symptoms, as well as the electrochemical skin conductance as a measure of sudomotor nerve function were chosen as outcomes. From all these parameters, the foot conductance and two questionnaires (Short form McGill Pain Questionnaire and Michigan Neuropathy Screening Instrument-questionnaire) reached the statistically significant mean in-between group difference in favor of the 20-week intervention diet.

## 4. Discussion

Lower DM prevalence among vegetarians compared to omnivores has been observed for decades, but it was well documented only recently, in particular through population studies of Seventh-Day Adventists. This religious movement originated in the United States and promotes a healthy lifestyle through the adherence to the vegetarianism with the avoidance of alcohol, tobacco, illegal drugs and caffeine-containing beverages. The rate of vegetarianism (including pesco-vegetarians and semi-vegetarians) in the Adventist Health study-2 was high; about fifty per cent and 4.48% of them were following vegan diets [12]: the prevalence of T2D in subjects aged 30 years or older was 2.6 × higher among nonvegetarians than in vegans. Different results were observed in an Indian population [13], where no significant reduction in DM prevalence among vegans was observed. There are some explanations: as an affiliation with veganism was self-reported, it is possible that the term “veganism” in the Indian context was not interpreted correctly. Indian vegans may include butter, ghee (a type of clarified butter derived from cow’s milk that contains fewer dairy proteins, but similar fat content than regular butter) or honey in their diet. Another important factor is the consumption of refined (white) rice: it is well documented that its higher consumption increases the rate of T2D [22]. Second, many Indian vegans do not practice this lifestyle due to personal health, environmental sustainability or political reasons. Third, there was no information reported regarding the type of DM. As the population was quite young (aged 20–49 years), we can presume that a number of participants who reported the presence of DM was affected by type 1 diabetes, which incidence is not correlated with the diet.

Neither in the study of Tonstad et al. [7] evaluating the incidence of DM in the same cohort of Adventist Health study-2, the type of diabetes was not specified, but we presume that the most of new cases was T2D since the mean age of subjects that reported DM was 62.5 years. Beside the vegan diet, also the lacto-ovo-vegetarian and semi-vegetarian (flexitarian) diets were protective against DM compared to non-vegetarians in non-Black participants (OR 0.684, 95% CI 0.542–0.862, OR 0.501, 95% CI 0.303–0.827, respectively). The semi-vegetarian diet was defined as a consumption of any type of meat one or more times per month, but less than once a week—we may equate it with a classic Mediterranean diet. Among Black subjects, only vegan and lacto-ovo-vegetarian diets were associated with a decreased risk of DM.

The major limitations of the included observational studies are that the diet assessments were self-reported and that in cross-sectional studies, the dietary habits may not be maintained in time. Nowadays, many individuals define themselves “vegans”, although they may include little animal product in their diet, especially after following the strict diet for a longer time [1]. Another point is that vegans may consume some foods that we do not consider healthy, such as chips, sweetened beverages or white rice. Obviously, the studies that dichotomously classified dietary patterns as vegan, various types of vegetarianism or omnivorous [7,11,12,13], may lead to a misclassification. On the contrary, several studies [8,10] investigated the variation in the degree of having a plant-based versus animal-based diet, considering the quality of plant foods consumed may not have this problem [8].

All randomized controlled studies reported the reduction in HbA_1c_ levels after the vegan diet, although the duration of the intervention varied from 6 to 74 weeks. Besides the length of follow-up, there are many other factors that make it difficult to compare the results among studies: duration of disease, medication regimens, baseline HbA_1c_ value, changes in medication during the intervention, baseline BMI, etc. It is worth noting that much of the effect of the intervention diets on glycemia appears to be mediated by weight reduction. Even more arduous is the comparison between the vegan and the omnivorous groups, as control diets in the included studies were not always neutral in energy balance and the approaches were different. In the vegan groups, mostly no restrictions in portion sizes, energy or carbohydrate intakes were placed; on the contrary, in some control groups, calorie-restricted diets were prescribed. These limitations are conclusive not only for the glycemic control parameters, but also for other cardiovascular risk factors (e.g., BMI, systolic and diastolic blood pressure, total cholesterol, HDL cholesterol, LDL cholesterol, triglycerides, etc.). We can presume that for some patients with T2D, the vegan diet may be an efficient way to lose weight and maintain glycemic control.

Besides the randomized controlled studies included in this systematic review, a recent meta-analysis of prospective cohort studies has also confirmed that the intake of plant-based proteins is significantly associated with a lower risk of cardiovascular disease mortality (pooled effect size 0.88, 95% CI 0.80–0.96) [23].

### 4.1. Potential Mechanisms

Despite the fact that the vegan diet is high in carbohydrates, all clinical trials included in this review have demonstrated its effect on glucose lowering, which is even stronger than that seen in other hypocaloric conventional diets recommended for people with DM [16,20]. This effect may be attributed to higher fiber content.

Dietary fibers lower the postprandial glucose response by well-known mechanisms such as the reduction in gastric emptying and consequent slower starch digestion and glucose absorption. Additionally, glucagon-like peptide 1 (GLP-1) plays an important role; besides a slower gastric emptying action, it improves glucose uptake and disposal in peripheral, especially in insulin-dependent tissues. GLP-1 reduces glucose production from the liver through the inhibition of glucagon secretion [24]. Some prospective cohort studies suggest that only cereal but not fruit or vegetable fiber intake is associated with reduced long-term risk for T2D [25,26].

Another important factor by which we may explain the benefits of the vegan diet on different cardiometabolic risk factors is the lower fat content. It has been demonstrated that fat-rich diets lead to an increased intramyocellular lipid (IMCL) concentration through the mitochondrial oxidative phosphorylation genes downregulation in skeletal muscle [27]. Excessive IMCL storage has a cytotoxic effect on mitochondria through reactive oxygen species (ROS) overproduction and increased metabolic stress which promotes the insulin resistance [28]. In fact, Goff et al. [29] have noticed a significantly minor content of IMCLs in the soleus muscle among 21 vegans compared to 25 omnivores, while IMCL contents in other muscle types were not significant (tibialis and gastrocnemius). Surprisingly, insulin sensitivity, expressed through the HOMA index (%S) was similar in both groups, albeit the fasting glucose level was significantly lower and beta-cell function (HOMA %B) significantly higher in vegans. Moreover, beyond the quantity, the type of fat consumed also has to be considered and it has to be stated that the connection between IMCLs and insulin sensitivity is not an absolute cause–effect relationship.

The improvement of insulin sensitivity was observed in correlation with lower serum ferritin levels several years ago [30], and later, it was well documented that the consumption of meat, especially red and processed, favors the development of T2D across increased insulin resistance [31]. As the vegan diet is absent of heme iron, we may suppose an enhancement of insulin sensitivity in vegans.

### 4.2. Potential Risk of Nutritional Deficiencies

It is well known that vegans are at risk of nutritional deficiencies, such as proteins, vitamin B12, calcium, vitamin D, iron, zinc or omega-3, if not well planned [32].

Many clinical studies have demonstrated that plant-based proteins have lower anabolic capacity compared to animal proteins due to their lower digestibility and higher limiting essential amino acid content (such as lysine in cereals or sulfur amino acids in legumes), which are more likely directed toward oxidation rather than being used for muscle protein synthesis. On the other hand, prospective cohort studies suggest that the difference in the anabolic effects of plant versus animal-based proteins could be reduced with an adequate protein intake (1.0–1.2 g of proteins/kg of body weight) or by blending different plant-based protein sources (e.g., cereals with legumes), which could mitigate sarcopenia, a common condition in elderly, especially in those with diabetes [33].

It is essential that all vegans get enough vitamin B12, as this micronutrient is generally found only in animal-sourced foods and its deficiency prevalence among vegans is high [34]. Nevertheless, vitamin B12 supplementation or the consumption of fortified foods among vegans is common, and often, its deficiency is not significantly higher than in omnivores [35]. An increased attention should be paid in subjects taking metformin, the first line pharmacotherapy in T2D, as it may lower B12 status [36].

Calcium and vitamin D are crucial for maintaining healthy bones and preventing osteoporosis, although the other nutrients listed above may also play a role. Vegans may be at higher risk of lower bone mineral density (BMD) and fractures due to an inadequate intake of these nutrients. Iguacel et al. [37] has confirmed the significantly lower total body BMD (mean difference—MD −0.059, 95% CI −0.106 to −0.012), lumbar spine BMD (MD −0.070, 95% CI −0.116 to −0.025) and femoral neck BMD (MD −0.055, 95% −0.090 to −0.021), as well as the significantly higher fracture risk (Risk Ratio (RR) 1.439, 95% CI 1.047–1.977) in vegans than in omnivores. As DM is associated with increased fracture risk [38], it is strongly recommended to encompass adequate calcium and vitamin D intake through diet or supplementation, as well as screen diabetic patients who follow a vegan diet for osteoporosis.

## 5. Conclusions

We can conclude that many large observational studies have demonstrated that the vegan diet is associated with lower T2D prevalence or incidence, although in some cohorts, it is not possible to distinguish if the beneficial effects derive from the vegan diet alone or from the overall healthy lifestyle. Furthermore, the results of randomized controlled studies performed in T2D patients have indicated its antihyperglycemic effect, even in the long-term.

Moreover, clinical trials with vegan diet in pre-diabetes and T2D people, in which the quantity of simple and complex carbohydrates and the quality of the nutrients taken will be evaluated, should be conducted. In this way, some unevenness of results obtained in the different reported clinical trials could be clarified and an appropriate vegan diet for pre-diabetic and T2D individuals could be created to be used as a nutraceutical intervention.

Finally, the vegan diet may be considered an acceptable and safe alternative to Western diets and seems to be comparable to other recommended healthful eating models (e.g., vegetarian, Mediterranean, Dietary Approaches to Stop Hypertension (DASH), etc.), but it must be stated that it has been considered to be a therapeutic diet with adverse effects associated with the long-term exclusion of some nutrients. Appropriate nutritional planning and surveillance conducted by dietitians and nutritionists trained in a vegan diet are recommended, as vegans are more likely to require vitamin B12 (especially who take metformin), vitamin D, calcium and iron supplementation, as well as a sufficient amount of protein. This is important, particularly in specific groups of diabetic patients such as frail elderly, adolescents, and pregnant and breastfeeding women.

Larger randomized controlled studies are necessary to confirm the effectiveness and safety of the vegan diet for diabetic patients.

## Figures and Tables

**Figure 1 nutrients-13-02123-f001:**
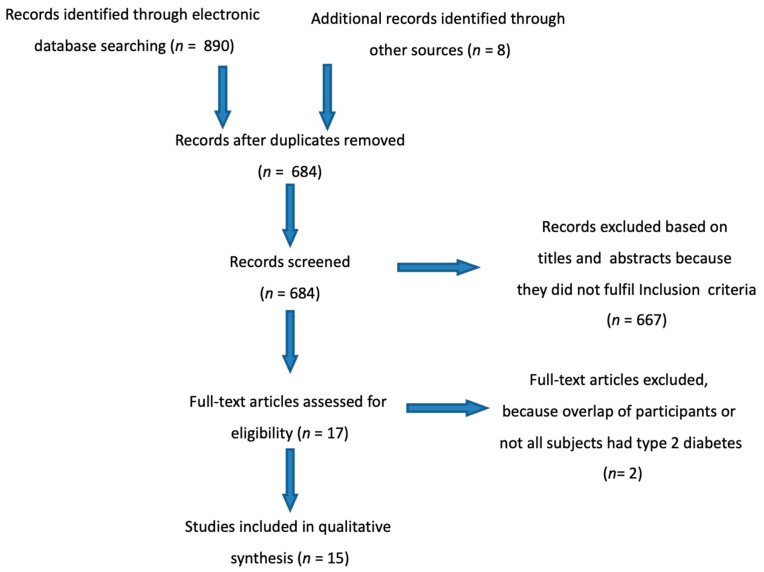
Flow diagram of study selection.

**Table 1 nutrients-13-02123-t001:** Characteristics of observational studies.

Reference	Country	Cohort	Sex	Follow-Up (Year)	Population	QA
Tonstad (2013) [7]	US and Canada	Adventist Health Study-2	M/F	2	41,387	fair
Satija (2015) [8]	US	Nurses Health Study + Nurses Health Study-2 + Health Professionals F–up Study	M/F	20	200,727	fair
Koloverou (2016) [9]	Greece	ATTICA Cohort Study	M/F	10	1485	good
Chen (2018) [10]	Netherland	Rotterdam Study	M/F	7.3	6798	good
Papier (2019) [11]	UK	Oxford–EPIC Study	M/F	17.6	45,314	good
Tonstad (2009) [12]	US and Canada	Adventist Health Study-2	M/F	Cross-sectional	60,903	fair
Agrawal (2014) [13]	India	National Family Health Survey-3	M/F	Cross-sectional	156,317	good

M = male, F = female, QA = quality assessment, EPIC = European Prospective Investigation into Cancer and Nutrition.

**Table 2 nutrients-13-02123-t002:** Characteristics of randomized controlled studies.

Reference	Country	Participants	Design	Follow-up (Week)	Control Diet	QA (Score)
Nicholson (1999) [14].	US	11 (7I + 4C)	P	12	Conventional low–fat diet	2.5
Wheeler (2002) [15].	US	17	CO	6	Animal–based protein diet	3
Barnard (2009) [16].	US	99 (49I + 50C)	P	74	Conventional diabetes diet	4
Ferdowsian (2010) [17].	US	19 (10I + 9C)	P	22	Usual diet	2.5
Mishra (2013) [18].	US	35 (17I + 18C)	P	18	Usual diet	4
Bunner (2015) [19].	US	33 (17I + 16C)	P	20	Usual diet	4
Lee (2016) [20].	Korea	93 (47I + 46C)	P	12	Conventional diabetes diet	3.5
Barnard (2018) [21].	US	40 (19I + 21C)	P	20	Portion-controlled diet	4

C = control, CO = crossover, I = intervention, P = parallel, QA = quality assessment.

**Table 3 nutrients-13-02123-t003:** Changes in HbA_1c_ levels (%) by group assignment.

Reference	Intervention Group				Control Group			Between-Group Difference (Intervention–Control) Mean (95% CI)	*p*-Value
	Baseline	Final	Change	Baseline		Final	Change		
Nicholson (1999) [14]	8.3 (1.7)	6.9 (1.1)	−1.4	8.0 (1.1)		7.0 (0.6)	−1.0		
Wheeler (2002) [15]	8.1 (0.4)	7.5 (0.3)	−0.6 **	7.9 (0.4)		7.4 (0.3)	−0.5 **	−0.1	0.75
Barnard (2009) [16]	8.05 (0.16)	7.71 (0.19)	−0.34 (0.19)	7.93 (0.14)		7.79 (0.18)	−0.14 (0.17)	−0.20 (−0.71 to 0.30)	0.43
Ferdowsian (2010) [17]	7.4 (0.3)	7.1 (0.5)	−0.3 (0.6)	7.0 (0.4)		6.7 (0.4)	−0.3 (0.2)	0 (−1.4 to 1.4)	0.97
Mishra (2013) [18]	7.52 (0.49)	6.78 (0.44)	−0.74 (0.19) **	7.03 (0.36)		7.13 (0.38)	0.1 (0.12)	−0.84 (−0.37 to −1.1) **	0.003
Bunner (2015) [19]	8.0 (1.7)	7.2 (1.4)	−0.8 (1.2) *	7.8 (1.6)		7.8 (1.4)	0.0 (0.9)	−0.7 (−1.5 to 0.1)	0.07
Lee (2016) [20]	7.7 (1.3)	7.1 (0.9)	−0.5 (0.8) **	7.4 (1.0)		7.2 (0.9)	−0.2 (0.7) *	−0.3 *	0.017
Barnard (2018) [21] median	6.7	6.2	−0.4 *	6.8		6.2	−0.4 **	0.1 (−0.2 to 0.6)	0.68

Data are presented as mean (SD, if available) unless otherwise indicated, * *p* < 0.05, ** *p* < 0.01, CI = confidence interval, SD = standard deviation.

## Data Availability

The data presented in this study are available on request from the corresponding author.

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
