# Peer review of "The Impact of Vegan Diet in the Prevention and Treatment of Type 2 Diabetes: A Systematic Review"

_nutrients, 2021, doi:10.3390/nu13062123_

Round 1
Reviewer 1 Report
The study is interesting and its design is well planned. However, the authors should make some changes to improve the presentation of the results obtained in the systematic review.
Materials and Methods
- Line 49: replace the word "through" with "untill"
- Line 67: A very important aspect in these studies is the self-definition that the subjects make of their diet. The authors of the different studies should report what methods or how the self-definition of a vegan diet is validated. Likewise, the authors of this review should make some reference to the way that this aspect has been controlled in each study.
Results:
Figure 2. This figure does not have to be in results. It refers to one of the cohorts; since the review covers several cohor studies, it does not make much sense to present the results in a figure of only one. The description only in the text, as the othe studies, is the most appropriate.
Discussion:
Line 377-379: "The rate of vegetarianism among Adventists is high, about fifty per cent. In the Adventist Health study – 2 [12], the prevalence of T2D in subjects aged 30 378 years or older was 2.6x higher among nonvegetarians than in vegans." Authors should specify the percentage of vegans, not the percentage of vegetarians in the Adventist study.
line 443-447: The authors should reconsider the sentence that one of the reasons for improved cardiometabolic risk is a low-fat diet. In this case they should point what consider a low-fat diet. Many dietary intake studies in vegans do not show "low-fat" diets. In addition, diets high in unsaturated fat have shown many cardiomatabolic benefits. This reason pointed out by the authors does not seem to have much sense.
Conclusions:
493: It is true that lifestyle factors influence the overall effect on the lower prevalence or incidence of DM, but the Adventist study treats these variables like all studies, as adjustment variables. The Adventist study is not the only one in which the vegetarian diet is associated with certain beneficial lifestyle variables. Therefore, the authors should remove "(e.g., Seventh - Day Adventists)."
505: what means "certified professionals"? health professionals trained in a vegan diet?
Author Response
Dear Editor,
It is with pleasure that we are resubmitting to your attention the article (nutrients-1056472) titled: "The impact of vegan diet in the prevention and treatment of type 2 diabetes: A systematic review" to be evaluated for a possible publication in your prestigious journal.
We are grateful to the reviewers for the insight comments, and we answer to them at the best of our abilities.
We are confident that after this revision the manuscript improved significantly.
We confirm that neither the manuscript nor any parts of its content are currently under consideration or published in another journal.
Thanks for your time and consideration.
Best Regards,
Davide Lauro
Reviewer 1
Materials and Methods: Line 49: replace the word "through" with "until" –
Answer: Done, as requested.
- Line 67: A very important aspect in these studies is the self-definition that the subjects make of their diet. The authors of the different studies should report what methods or how the self-definition of a vegan diet is validated. Likewise, the authors of this review should make some reference to the way that this aspect has been controlled in each study.
Answer: As described in lines 390-391: “The major limitations of included observational studies are that the diet assessments were self-reported.
Results: Figure 2. This figure does not have to be in results. It refers to one of the cohorts; since the review covers several cohor studies, it does not make much sense to present the results in a figure of only one. The description only in the text, as the othe studies, is the most appropriate.
Answer: Thank-you for the suggestion. Figure 2 has been removed and the prevalence of T2D in each class of vegetarianism was described in the text, lines 147-148.
Discussion: Line 377-379 (now 367-368): "The rate of vegetarianism among Adventists is high, about fifty per cent. In the Adventist Health study – 2 [12], the prevalence of T2D in subjects aged 30 378 years or older was 2.6x higher among nonvegetarians than in vegans." Authors should specify the percentage of vegans, not the percentage of vegetarians in the Adventist study.
Answer: Thank-you for the comment. Now the information that “the rate of vegetarianism among Adventists is about 50%” is general. We have added the information about the rate of veganism in the Adventist Health study – 2 cohort, line 367.
line 443-447: The authors should reconsider the sentence that one of the reasons for improved cardiometabolic risk is a low-fat diet. In this case they should point what consider a low-fat diet. Many dietary intake studies in vegans do not show "low-fat" diets. In addition, diets high in unsaturated fat have shown many cardiomatabolic benefits. This reason pointed out by the authors does not seem to have much sense. Answer: Generally, the vegan diet is considered a low – fat diet.
Conclusions: 493(now 482): It is true that lifestyle factors influence the overall effect on the lower prevalence or incidence of DM, but the Adventist study treats these variables like all studies, as adjustment variables. The Adventist study is not the only one in which the vegetarian diet is associated with certain beneficial lifestyle variables. Therefore, the authors should remove "(e.g., Seventh - Day Adventists)."
Answer: We have removed the text in brackets.
505 (now 495): what means "certified professionals"? health professionals trained in a vegan diet?
Answer: Than-you again for the appropriate comment. We have corrected, as requested and we change the sentence in “dietitians and nutritionists trained in a vegan diet”, lines 494-495.
Please find the modified manuscript in attachment.
Reviewer 2 Report
The authors present a re-submitted systematic review on observational and interventional studies, assessing the metabolic potential of vegan diets with respect to T2D and its associated parameters.
The introduction (also in the abstract) has been improved in comparison to the previous version. The nature of long-term benefits is now described correctly on the basis of cohort studies, only.
Methods: It is unclear, why the authors chose to conduct a sole review over a full SRMA. Are there statistical reasons, which do not allow for calculation of forest plots? Please clarify in the manuscript.
Results: This section is still not very comprehensive and easy to read. Using the tables in the actual results section would support the presentation of the synthesis of common results. All tables should, however, include only the same parameter. Glucose AUCs in the fasting glucose section are not useful, the same applies to TG AUCs for TGs and GFR, RPF or AER in the albuminuria table.
Limitations: Typical problems of cohort studies are even more aggravated as the vegan lifestyle is strongly connected to religious beliefs or other general lifestyle choices, affecting physical activity, smoking, drinking as additional major risk factors for T2DM and metabolic outcomes.
Discussion and conclusions: The conclusions are mostly well balanced and clearly distinguish epidemiological and interventional evidence. The statement of an "acceptable and safe alternative" (L. 501) is debatable, as the authors themselves address the risks of malnutrition and the overall lack of effectiveness and safety data from larger RCTs. The meta-analyses by Neuenschwander and Schwingshackl et al. (all 2018-2019) indicate, that the vegan diet is poorly investigated and - as far as published - ranks considerably lower in the improvement of glycemia, lipid profile, blood pressure and body weight, when compared to MedDiet, DASH or low-carb.
Author Response
Dear Editor,
It is with pleasure that we are resubmitting to your attention the article (nutrients-1056472) titled: "The impact of vegan diet in the prevention and treatment of type 2 diabetes: A systematic review" to be evaluated for a possible publication in your prestigious journal.
We are grateful to the reviewers for the insight comments, and we answer to them at the best of our abilities.
We are confident that after this revision the manuscript improved significantly.
We confirm that neither the manuscript nor any parts of its content are currently under consideration or published in another journal.
Thanks for your time and consideration.
Best Regards,
Davide Lauro
Reviewer 2
The introduction (also in the abstract) has been improved in comparison to the previous version. The nature of long-term benefits is now described correctly on the basis of cohort studies, only.
Methods: It is unclear, why the authors chose to conduct a sole review over a full SRMA. Are there statistical reasons, which do not allow for calculation of forest plots? Please clarify in the manuscript.
Answer: Thank-you for your comments, however, as we clarified in the previous cover letter, there are no statistical reasons which do not allow for calculation of forest plots, we wanted to conduct a sole review, not a full SRMA, to comment more aspects of this topic. We consider it is not necessary to clarify this aspect in the text.
Results: This section is still not very comprehensive and easy to read. Using the tables in the actual results section would support the presentation of the synthesis of common results. All tables should, however, include only the same parameter. Glucose AUCs in the fasting glucose section are not useful, the same applies to TG AUCs for TGs and GFR, RPF or AER in the albuminuria table.
Answer: Thank-you for your comments, anyways, we would like to present the tables of each of variables only in the supplementary section, since the article would be too long otherwise. We decided to include different units of the same parameters to demonstrate the trend of such parameter, considering it is not necessary to calculate the forest plots.
Limitations: Typical problems of cohort studies are even more aggravated as the vegan lifestyle is strongly connected to religious beliefs or other general lifestyle choices, affecting physical activity, smoking, drinking as additional major risk factors for T2DM and metabolic outcomes.
Answer: As described in lines 479 – 482: “We can conclude that many large observational studies have demonstrated that the vegan diet is associated with lower T2D prevalence or incidence, although in some cohorts it is not possible to distinguish if the beneficial effects derive from the vegan diet alone or from the overall healthy lifestyle.”
Discussion and conclusions: The conclusions are mostly well balanced and clearly distinguish epidemiological and interventional evidence. The statement of an "acceptable and safe alternative" (L. 501, now line 490) is debatable, as the authors themselves address the risks of malnutrition and the overall lack of effectiveness and safety data from larger RCTs. The meta-analyses by Neuenschwander and Schwingshackl et al. (all 2018-2019) indicate, that the vegan diet is poorly investigated and - as far as published - ranks considerably lower in the improvement of glycemia, lipid profile, blood pressure and body weight, when compared to MedDiet, DASH or low-carb.
Answer: Thank-you for your suggestion. We think to have addressed since it is reported in line 490:“...the vegan diet is an acceptable and safe alternative to western diets..”, but it is also stated in lines 493 – 495: “Appropriate nutritional planning and surveillance conducted by dietitians and nutritionists trained in a vegan diet (further modified) are recommended. we aimed to send the message that vegan diet seems to be healthier compared to western diet, but its use should be “conscious” and the vegans should be guided by health professionals. As suggested, we added the sentence in the conclusions (lines 500 - 501) as suggested.
Please find the modified manuscript in attachment
Round 2
Reviewer 2 Report
The revision was successful.
Author Response
Thank you.
This manuscript is a resubmission of an earlier submission. The following is a list of the peer review reports and author responses from that submission.
Round 1
Reviewer 1 Report
The authors present a systematic review on observational and interventional studies, assessing the metabolic potential of vegan diets with respect to T2D and its associated parameters.
The introduction (also in the abstract) is too optimistic; benefits for cancer and obesity incidence were shown in cohort studies, only. Therefore, an "effect" (i.e. causality) cannot be deducted. Similarly, the discussion of the results from observational studies should not be titled with "primary prevention", as prevention was not assessed. Cohort studies are unable to show a preventive effect, rather than just showing an association of unclear causality.
Thus, the conclusion of being an "efficient tool for prevention" is misleading. (L. 480)
Methods: It is unclear, why the authors chose to conduct a sole review over a full SRMA. Are there statistical reasons, which do not allow for calculation of forest plots?
Results: In the current form, the results section is not very comprehensive and, thus, hard to read. The synthesis of common results is not apparent.
I recommend to use more tables for a proper overview regarding metabolic outcomes and effects with significant superiority (!) over standard diet.
Discussion: Mechanistic consideration of dietary fiber is certainly correct; however, only insoluble (cereal) fiber are associated with reduced long-term risks including T2D risk (Schulze et al. 2007). Soluble fiber does not appear to be protective in cohort studies (Interact; 2015), and does not show consistent interventional benefits (Reynolds et al. 2020). Insoluble cereal may be the crucial component (Barber et al. 2020; Honsek et al. 2018).
The correlation between (total) fat intake, metabolic state and also IMCL content is mislieading. Fat quality is clearly more important, and IMCL content is not uniquely connected to fat intake, but several other factors.
Conclusions: The data summarised do NOT support vegan diets as an unanimously safe diet (l. 485; see chapter on health risk and nutritional deficiencies). Apart from effects on HbA1c, there is NO evidence for reduced risk of complications (l. 485). Recent network meta-analyses (Neuenschwander et al., Schwingshackl et al.) rated vegetarian diets as moderately effective diets, and low-fat diets as one of the weekest options for T2D treatment at all. (l. 486 f.)
Reviewer 2 Report
The study is interesting and necessary since there are no specific systematic reviews on the relationship between DM2 and the vegan diet. However, the information provided in the article does not allow its publication. Data from the different observational studies and RCTs synthesized and with statistical analysis of these data are necessary. Similarly, when presenting the data from the studies, it is necessary to consider other factors that affect the incidence and management of DM2, such as body weight, level of physical activity, etc. Authors should re write the article introducing all the data obtained from this systematic review and then submit the article again for publication.Reviewer 3 Report
Thank you for the opportunity to review this manuscript, which is a systematic review of a vegan or plant-based diet and risk of type 2 diabetes or outcomes related to glycemic control and diabetes complication. The authors included both observational studies and dietary intervention trials in this review and concluded the overall benefit of a vegan diet for type 2 diabetes prevention and management. I have major comments on: 1) the scientific merit of publishing this systematic review; 2) the systematic review method; and 3) the authors’ conclusions, as well as minor specific comments as below.
Major comments:
1) The scientific merit of publishing this systematic review, in light of recently published systematic review and meta-analysis articles on similar topics
Systematic review articles on very similar topics have been published recently and the scientific merit/significance of this manuscript is uncertain. The authors did not mention how their systematic review is distinct from these existing systematic reviews on very similar topics and provides new information. In addition, these recent publications include meta-analyses; however, this manuscript does not. Based on these publications, it appears possible to conduct a meta-analysis. The authors need to justify the reason for not conducting a meta-analysis as well as the merit and novelty of this manuscript.
2) The systematic review method - inclusion of vegan and plant-based diets, but not vegetarian or other terminology in the search terms; and the selection of studies
First, search terms on diets are vegan and plant-based diets, which do not necessarily cover vegetarian diets that are not vegan. Hence, this search strategy has likely led to missing many studies that should have been included in this systematic review. In addition, the commonly accepted definition of the plant-based diet does not necessarily exclude all foods of animal origin and may include meats. The authors’ definition (Lines 58-59) seems to be synonymous with vegan diet and it would be helpful to use more commonly accepted definition. This likely has resulted incomplete search of the existing literature for the topic. Second, cross-sectional studies are not one type of cohort studies. By study design, cohort studies assess incidence, whereas cross-sectional studies assess prevalence (lines 173-174). In RCTs, it is understood that the diagnosis of diabetes is not an outcome and hence the authors included a variety of outcomes that are related to glycemic control and diabetes complications (lines 61-67); however, none of these outcomes included from RCTs are part of the search terms (lines 51-54). For example, body mass index does not necessarily show up in search when diabetes is the only search term included. This might have resulted in missing many studies that address your research question. In addition, it is very difficult to follow findings from studies identified in systematic review and please include study findings in tables, not only describing in text.
3) The authors’ conclusions not clearly linked with study findings
The authors' conclusions do not appear to be solely based on findings they summarized from studies identified in the systematic review and appear to conflict with some of the findings. For example, conclusions mentioned in lines 482-485 do not agree with findings mentioned in lines 315-318. Although limited information is included, lines 315-318 read that not all findings support the conclusion mentioned in lines 482-485. For instance, one study reported an increase in LDL (lines 315-318). The authors need to write conclusions based on the findings.
Minor specific comments:
Lines 136-138: Which risk of bias assessment was used for cross-sectional studies? They are not cohort studies
Line 239: Please change to “dietary”, instead of “food” supplement use.
Lines 248-240: This sentence does not sound logical and please re-write and clarify.
Lines 350-352: Please clarify the existence of (or lack thereof) statistical significant difference in this sentence.
Lines 324-332: Please cite all studies that reported triglyceride levels and clarify the significant change was increase or reduction.
Lines 393-395: Please include the reference(s) for this statement.
Lines 400-401: Please clarify and base your opinion on your findings from the systematic review.
Lines 451-477: This section is informative; however, it does not put your findings into context. Given this manuscript is not a guideline on vegan diets for diabetes, this section needs to cite and discuss studies included in the systematic review. What are levels of each nutrient of concern listed for vegan diets in each study? Were any of other health outcomes such as fracture risk reported in these studies?
Line 463: Please clarify whether it is assume or consume.
Line 466 and 496: Please clarify whether it is assuming/assume or taking/take metformin.